# Caffeic Acid Esters Are Effective Bactericidal Compounds Against *Paenibacillus*
*larvae* by Altering Intracellular Oxidant and Antioxidant Levels

**DOI:** 10.3390/biom9080312

**Published:** 2019-07-27

**Authors:** William Collins, Noah Lowen, David J. Blake

**Affiliations:** 1Department of Biochemistry and Chemistry, Fort Lewis College, 1000 Rim Dr., Durango, CO 81301, USA; 2Department of Biology, Fort Lewis College, 1000 Rim Dr., Durango, CO 81301, USA

**Keywords:** *Apis millifera*, *Paenibacillus larvae*, propolis, caffeic acid esters, minimum inhibitory concentration (MIC)

## Abstract

American Foulbrood (AFB) is a deadly bacterial disease affecting pupal and larval honey bees. AFB is caused by the endospore-forming bacterium *Paenibacillus larvae* (PL). Propolis, which contains a variety of organic compounds, is a product of bee foraging and is a resinous substance derived from botanical substances found primarily in trees. Several compounds from the class of caffeic acid esters, which are commonly found in propolis, have been shown to have antibacterial activity against PL. In this study, six different caffeic acid esters were synthesized, purified, spectroscopically analyzed, and tested for their activity against PL to determine the minimum inhibitory concentrations (MICs) and minimum bactericidal concentrations (MBCs). Caffeic acid isopropenyl ester (CAIE), caffeic acid benzyl ester (CABE), and caffeic acid phenethyl ester (CAPE) were the most effective in inhibiting PL growth and killing PL cell with MICs and MBCs of 125 µg/mL when used individually, and a MIC and MBC of 31.25 µg/mL for each compound alone when CAIE, CABE, and CAPE are used in combination against PL. These compounds inhibited bacterial growth through a bactericidal effect, which revealed cell killing but no lysis of PL cells after 18 h. Incubation with CAIE, CABE, and CAPE at their MICs significantly increased reactive oxygen species levels and significantly changed glutathione levels within PL cells. Caffeic acid esters are potent bactericidal compounds against PL and eliminate bacterial growth through an oxidative stress mechanism.

## 1. Introduction

USA and European beekeepers face high honey bee (*Apis millifera*) colony losses each year. Indeed, the BeeInformed Partnership Survey, a national epidemiological analysis of honey bee health, indicated that 40.1% of all USA colonies were lost in 2018 [1]. There are many factors involved in these losses and it is now generally accepted that no single issue is completely responsible for the colony mortality increases observed over the last 10 years [2]. However, viruses, bacteria, fungi, and parasites that commonly harm honey bees combined with the chemical inputs routinely utilized to ameliorate these afflictions are now understood to be a major cause of the decline of honey bee health worldwide [3,4]. 

One of the most dangerous pathogens that can infect honey bee colonies is the endospore-forming bacterium *Paenibacillus larvae* (PL) [5]. This extremely contagious bacterium, known commonly in the beekeeping community as American foulbrood (AFB), adversely affects honey bees during the larval or pupal stages [5,6]. The endospore, once inside a colony, is typically distributed to larvae during routine feeding by nurse bees, and after ingestion of the spores ultimately results in septicemia of the larvae [7]. The ensuing corpses then release PL spores, which are concomitantly dispersed throughout the brood nest [6,8,9]. Colony mortality is commonly observed within one season after infection, and infected colonies can then horizontally transmit PL spores to other colonies [10].

While a colony suffering from AFB can be identified by conspicuous symptoms such as irregular brood cappings, dark and punctured caps emitting foul odors, and brown residues of dead larvae [11], few treatment options exist to effectively fight the infection. Currently, broad spectrum antibiotics such as sulfathiazole, tylosin tartrate/Tylan, and oxytetracycline hydrochloride (OTC) are used to inhibit the growth of PL. However, the extreme dependence on these therapies and their subsequent overuses has led to the appearance of resistant strains [12]. As a result, the use of antibiotics to treat AFB in many countries (including all of the EU members) is now illegal, and as a direct consequence, colonies infected with PL in these countries must be destroyed through burning to prevent further spread of the disease [13]. Compounding this issue, as of 2017, the Food and Drug Administration now requires beekeepers within the USA to get a prescription from a veterinarian each time they use antibiotics within a honey bee colony to combat AFB [14].

Understandably, honey bees already possess a variety of social immune responses that attempt to mitigate colony-level infections such as AFB [15,16,17]. The hot, humid, and densely packed interior of a bee colony is an ideal environment for the propagation of many pathogens [18]. Therefore, to prevent unwanted pathogenic growth inside colonies, bees actively gather phytochemical-rich plant resins (termed propolis when mixed with wax) and deposit them throughout the structure of the colony [19,20,21]. The phytochemical composition of these resins is a complex mixture of biologically active terpenes, terpenoids, and phenols, with as many as 300 different molecules found within each resin sample [22,23,24]. Importantly, recent research has shown some propolis extracts to have biological activity against PL [25].

Unfortunately, because the chemical composition of propolis varies by location, season, and botanical availability it is difficult to standardize the antimicrobial efficacy of propolis extracts across multiple studies [26,27,28]. Additionally, the chemical variability of propolis (and thus efficacy) also creates considerable challenges in the development of quality-controlled therapies [29,30]. To directly address this issue, our laboratories have begun a program in the systematic synthesis and testing of individual natural product components of propolis against PL. We have begun our investigations with the caffeic ester family of molecules. Members of this class of molecules have already been shown to have considerable biological activity against many different Gram-positive bacteria and are also commonly identified in North American and European propolis samples found near poplar resin sources [22,31,32]. Moreover, extracts of mixtures of caffeic esters have been shown to inhibit PL growth [33]. The aim of this work is to show that, using compounds inexpensively prepared in a laboratory on a gram scale, we are able to not only determine potential highly effective compounds for critically needed AFB therapies, but also a putative mechanism of antibiotic action for this class of molecules.

## 2. Materials and Methods 

### 2.1. General Materials for Synthetic Procedures

Reagents and solvents were purchased from commercial sources and used without further purification unless otherwise specified. Reaction solvents, and solvents for liquid phase extraction were: ethyl acetate (Fisher Scientific, Suwanee, GA, USA, ACS grade,), hexanes (Acros, Trenton, NJ, USA, ACS grade), dichloromethane (Fisher Scientific, ACS grade), methanol (Acros, ACS grade), ethanol (Acros, ACS grade), dimethylformamide (Aldrich, Raleigh, NC, USA, ACS grade), and toluene (Acros, ACS grade). Reagents used were: oxalyl chloride (CAS: 79-37-8, Sigma, Allentown, PA, USA, ACS grade), pyridine (CAS: 110-86-1, Aldrich, ACS grade), benzylalcohol (CAS: 100-51-6, Aldrich, reagent plus grade), cinnamyl alcohol (CAS: 104-54-1, Aldrich), phenethylalcohol (CAS: 60-12-8, Aldrich), 3-methyl-2-butenyl alcohol (CAS: 556-82-1, Aldrich), magnesium sulfate (CAS: 7487-88-9, Aldrich), potassium carbonate (CAS: 584-08-7, Aldrich), caffeic acid (CAS: 331-39-5, Aldrich), and ferulic acid (CAS: 1135-24-6, Aldrich). TLC/chromatography solvents used were: hexanes (CAS: 110-54-3, Fisher Scientific, ACS grade), and ethyl acetate (CAS: 141-78-6, Fisher Scientific, ACS grade). Diacetylcaffeic acid was prepared from caffeine acid in near quantitative yield (98%) and on a gram scale by following a known protocol (LeBlanc et al. 2012). ^1^H and ^13^C NMR was recorded on JEOL (400 MHz ^1^H spectrometer and 125 MHz ^13^C spectrometer) in deuterochloroform (CAS: 865-49-6, Aldrich, ACS grade) or DMSO-d_6_ (CAS: 2206-27-1, Aldrich, ACS grade). Spectra were referenced to residual chloroform (7.26 ppm, ^1^H). Chemical shifts are reported in ppm, multiplicities are indicated by s (singlet), d (doublet), t (triplet), q (quartet), p (pentet), h (hextet), m (multiplet), and br (broad). Coupling constants, J, are reported in Hertz. Infrared spectra (IR) were recorded on a Thermo Fisher Scientific Nicolet iS10 spectrophotometer. Peaks are recorded in cm^−1^ with indicated relative intensities: s (strong, 67–100%); m (medium, 34–66%); and w (weak, 0–33%). Analytical thin-layer chromatography was performed on Merck (Kenilworth, NJ, USA) silica gel 60 plates with F-254 indicator. Visualization was accomplished by ultraviolet (UV) light. Silica gel used for column chromatography was obtained from SiliCycle. All yields refer to spectroscopically (^1^H-NMR) homogeneous material.

### 2.2. Preparation of Caffeic Ester Derivatives

General procedure for esterification: Diacetylcaffeic acid (580 mg, 2.195 mmol, 1 eq.) was dissolved in CH_2_Cl_2_ (12 mL, 188 mmol) and placed under a nitrogen atmosphere. Oxalyl chloride (0.38 mL, 4.39 mmol, 2 eq.) was added dropwise to this solution before adding DMF (5 drops). The reaction mixture was stirred for 16 h at room temperature. Excess oxalyl chloride was removed under reduced pressure via rotovap, and the resulting oil was dissolved in toluene (25 mL). Pyridine (2 mL) was added dropwise to this solution before adding the alcohol undergoing the esterification reaction (4.4 mmol, 2 eq.). This solution was stirred at room temperature for 3 h. The solvent was then removed via rotovap, and the resulting oil was dissolved in EtOAc (50 mL) before being transferred to a separatory funnel. The aqueous layer was removed, and the organic layer was washed with deionized water (2 × 50 mL) followed by brine (2 × 50 mL). The organic layer was dried over MgSO_4_, filtered, and concentrated on a rotovap. The crude residue was purified by column chromatography (silica gel, EtOAc/hexanes gradient: 20/80 to 80/20). 

#### 2.2.1. Caffeic Acid Cinnamyl Ester Diacetate

White solid (73%). IR (ATR) cm^−1^: 1762 (s), 1704 (s), 1635 (w), 1506 (w), 1369 (w), 1317 (w), 1256 (m); ^1^H NMR (CDCl_3_) 7.19–7.43 (m, 5H), 7.65 (d, J = 16.5 Hz, 1H), 7.40 (d, J = 7.3 Hz, 1H), 7.35 (d, J = 1.4 Hz, 1H), 7.32 (d, J = 7.3 Hz, 1H), 6.68 (d, J = 16.0 Hz, 1H), 6.41 (d, J = 16.0 Hz, 1H), 6.32 (dt, J = 16.0, 6.9 Hz, 1H), 4.85 (t, J = 6.9 Hz, 1H), 2.3 (apd, 6H); ^13^C NMR (CDCl_3_) 168.3, 168.1, 165.6, 143.9, 143.5, 142.9, 136.2, 135.4, 133.3, 128.9, 128.5, 126.9, 126.7, 124.1, 123.1, 122.9, 65.2, 20.5, 20.5.

#### 2.2.2. Caffeic Acid Benzyl Ester Diacetate

White solid (75%). IR (ATR) cm^−1^: 3347 (w), 3308 (w), 1687 (m), 1598 (m), 1535 (w), 1444 (w), 1364 (w), 1278 (m), 1172 (m); ^1^H NMR (CDCl_3_) 7.67–7.63 (d, J = 16.0 Hz, 1H), 7.39 (s, 1H), 7.38 (d, J = 11.9 Hz, 1H), 7.22–7.19 (d, J = 8.2 Hz, 1H), 6.44 (d, J = 16.0 Hz, 1H), 5.23 (s, 1H), 2.29–2.28 (apd, 6H); ^13^C NMR (CDCl_3_) 168.1, 168.0, 166.4, 143.6, 143.3, 142.5, 135.9, 133.3, 128.7, 128.3, 126.4, 124.0, 122.8, 119.1, 66, 20.7, 20.0.

#### 2.2.3. Caffeic Acid Phenethyl Ester Diacetate

White solid (89%). IR (ATR) cm^−1^: 3381 (w), 3329 (w), 3066 (w), 3030 (w), 2942 (w), 2895 (w), 2870 (w), 1766 (s), 1697 (s), 1636 (s), 1584 (w), 1507 (s), 1455 (w), 1420 (s), 1369 (s), 1321 (s), 1276 (s), 897 (s); ^1^H NMR (CDCl_3_) 7.6 (d, J = 16.0, 1H), 7.4 (d, J = 1.8, 1H), 6.4 (d, J = 16.0, 1H), 4.4 (t, J = 2.1, 2H), 3.0 (t, J = 2.1, 2H), 2.3 (d, J = 3.7, 6H); ^13^C NMR (CDCl_3_) 168.2, 168.1, 166.6, 143.6, 143.0, 142.5, 137.9, 133.4, 129.0, 128.6, 126.7, 126.5, 124.0, 122.8, 119.3, 65.3, 35.3, 20.8.

#### 2.2.4. Caffeic Acid Isopropenyl Ester Diacetate

White solid (41%). IR (ATR) cm^−1^: 3380 (w), 3331 (w), 2940 (w), 2895 (w), 1760 (w), 1699 (s), 1507 (s), 1450 (w), 1420 (s), 1276 (s); ^1^H NMR (CDCl_3_) 7.62-7.58 (d, J = 16.0 Hz, 1 H), 7.39-7.37 (m, J = 8.2, 2.3 Hz, 1H), 7.33–7.33 (d, J = 2.3 Hz, 1 H), 7.21–7.19 (d, J = 8.2 Hz, 1H), 6.39-6.35 (d, J = 16.0 Hz, 1 H), 5.42–5.38 (t, J = 8.7, 7.3 Hz, 1 H), 4.70–4.68 (d, J = 7.3, 2 H), 2.29-2.29 (d, J = 2.6 Hz, 6H) 1.77–1.74 (d, J = 15.1, 6 H) ^13^C NMR (CDCl_3_) 171.0, 168.0, 166.0, 166.2, 143.7, 142.5, 139.2, 133.1, 126.3, 124.3, 122.8, 119.4, 118.7, 61.5, 60.3, 26.1, 20.5, 17.8, 14.2

General procedure for deprotection: To a round bottom flask was added the purified acetylated ester (3.24 mmol) and MeOH:CH_2_Cl_2_ (12 mL, 1:1). To this solution was added K_2_CO_3_ (330 mg, 3.53 mmol) and the reaction was stirred at room temperature. The reaction was monitored for completion via TLC (1:1 EtOAc: hexanes) and, once complete, the solvent was removed under reduced pressure in a rotovap. The resulting residue was dissolved in EtOAc (50 mL), and the aqueous layer was removed. The organic layer was washed with water (30 mL) and brine (30 mL) before being dried over MgSO_4_, filtered, and concentrated via rotovap. 

#### 2.2.5. Caffeic Acid Cinnamyl Ester (CACE)

Brown solid (68%). IR (ATR) cm^−1^: 3347 (w), 3308 (w), 1687 (m), 1598 (m); ^1^H NMR (DMSO) 7.57 (d, J = 16.5 Hz, 1H), 7.38 (d, J = 7.3 Hz, 2H), 7.32–7.18 (m, 3H), 7.06 (d, J = 0.9 Hz, 1H), 6.91 (d, J = 8.2 Hz, 1H), 6.86 (d, J = 15.6 Hz, 1H), 6.65 (d, J = 0.9 Hz, 1H), 6.32 (dt, J = 15.7, 6.9 Hz, 1H), 6.26 (d, J = 16.0 Hz, 1H), 4.81 (t, J = 6.9 Hz, 1H); ^13^C NMR (DMSO) 167.2, 147.3, 145.5, 144.8, 136.5, 134.1, 128.9, 128.2, 126.9, 123.8, 122.1, 115.6, 114.9, 114.3, 65.1

#### 2.2.6. Caffeic Acid Benzyl Ester 

Tan solid (33%). IR (ATR) cm^−1^: 3459 (m), 3319 (m); 1690 (m), 1602 (m); ^1^H NMR (DMSO) 7.25–7.21 (d, J = 15.6 Hz, 1H), 7.2 (s, 1H), 7.01 (s, 1H), 6.68 (s, 1H), 6.54–6.44 (m, 2H), 5.89-5.85 (d, J = 16.0 Hz, 1H), 4.85 (s, 1H), 3.72–3.71 (d, J = 6.9 Hz, 1 H), 1.65 (s, —OH), 0.89 (s, —OH); ^13^C NMR (DMSO) 167.0, 148.1, 145.5, 145.3, 136.2, 128.4, 127.9, 126.0, 121.5, 115.6, 114.4, 113.9, 65.7

#### 2.2.7. Caffeic Acid Phenethyl Ester 

Light brown solid (79%). IR (ATR) cm^−1^: 3065 (w), 3031 (w), 2966 (w), 2894 (w), 2871 (w), 1767 (s), 1697 (s), 1602 (w), 1582 (w), 1507 (s), 1454 (w), 1422 (m), 1369 (s), 1321 (m), 1276 (m), 1256 (s), 1107 (s), 989 (s), 874 (s), 701 (s); ^1^H NMR (DMSO) 7.5 (d, J = 16.0, 1H), 7.0 (s, 1H), 6.9 (d, J = 10.8, 1H), 6.8 (d, J = 7.8, 1H), 6.2 (d, J = 15.6, 1H), 4.4 (t, J = 6.9, 2H), 3.0 (t, J = 6.9, 2H); ^13^C NMR (DMSO) 167.3, 147.5, 145.3, 145.0, 138.0, 128.9, 128.4, 126.7, 126.5, 121.8, 115.5, 114.6, 114.3, 64.7, 35.2 

#### 2.2.8. Caffeic Acid Isopropenyl Ester

Light brown solid (61%). IR (ATR) cm^−1^: 3060 (w), 2961 (w), 2871 (w), 1767 (s), 1697 (s), 1605 (w), 1579 (w), 1510 (s), 1454 (w), 1422 (m), 1275 (m), 872 (s); ^1^H NMR (DMSO) 7.56(d, J = 16.0 Hz, 1H), 7.09 (s, 1H), 6.99 (d, J = 8.7 Hz, 1H), 6.85 (d, J = 8.2 Hz, 1H), 6.25 (d, J = 16.0, 1H), 5.4 (t, J = 6.5 Hz, 1H), 4.67 (d, J = 7.3 Hz, 2H), 1.75 (apd, J = 14.7, 6H); ^13^C NMR (DMSO) 168.5, 146.7, 145.6, 144.0, 139.7, 127.3, 122.4, 118.4, 115.6, 115.2, 114.5, 61.8, 25.9, 18.0

### 2.3. Bacterial Strains and Growth Conditions

*Paenibacillus larvae* (PL) was obtained from ATCC #9545 (Manassas, VA, USA) and cultured in either brain-heart infusion medium supplemented with 1 mg/L thiamine (BHIT) or Mueller-Hinton broth, yeast extract, potassium phosphate, glucose, and pyruvate (MYPGP) supplemented with 1 mg/L thiamine at 37 °C as previously described [34]. The strain was maintained in BHIT and new cultures were inoculated every 24–48 h. The day before each assay described below, an overnight agitated culture was prepared in BHIT or MYPGP liquid broth and set in a 37 °C incubator for an overnight incubation. BHIT was used to culture *P. larvae* as suggested by the manufacture (ATCC), and MYPGP was used because is routinely used to cultivate *P. larvae* for AFB diagnosis and yields the highest percentage of spore recovery [34].

### 2.4. Determination of Minimal Inhibitory Concentration and Minimal Bactericidal Concentration

Stock concentrations of each compound were freshly prepared as 10.5 mg/mL solutions in dimethyl sulfoxide (DMSO). Each stock solution was used for serial dilutions in a 96-well microplate from 0 to 500 µg/mL (final concentration) including 250, 125, 62.5, 31.25, 15.63, 7.81, and 3.90 µg/mL concentrations as previously described [33]. Briefly, 10 µL of an overnight bacterial suspension was added to each well containing 180 µL of either BHIT or MYPGP. Bacterial growth in either BHIT or MYPGP did not affect the minimum inhibitory concentration (MIC). 10 µL of the serially diluted compounds was added to each well. The microplate was incubated for 18 h at 37 °C. Absorbance was acquired at 600 nm using the Infinite M200 Microplate reader (Tecan, San Jose, CA, USA). For the negative control, *P. larvae* cells were prepared with culture media, bacteria, and the corresponding amount of DMSO only. For the positive control, 10 µL of tetracycline hydrochloride (Fisher Scientific, Suwanee, GA) was added to each well at a final concentration of 20 µg/mL. Four independent biological replicates were performed to establish the MIC using two to six technical replicates for each experiment (*n* = 2–6). For the minimum bactericidal concentrations (MBC) assay, serially diluted bacterial samples from the 18 h MIC experiments were plated on BHIT agar plates and incubated for an additional 18 h overnight at 37 °C. Colonies were enumerated and the lowest compound concentration required to achieve bactericidal killing, which was defined as a 99.9% reduction in the initial inoculum, was identified as the MBC.

### 2.5. Flow Cytometry Analysis

To determine whether the inhibitory caffeic acid derivatives were bacteriostatic or bacteriolyticat their MICs, flow cytometry was used as previously described [35]. Briefly, *P. larvae* were incubated overnight in BHIT with either caffeic acid isopropenyl ester (CAIE), caffeic acid benzyl ester (CABE), or caffeic acid phenethyl ester (CAPE) at 125 µg/mL for 18 h at 37 °C. After incubation, 50 µL of each bacterial sample was diluted in 425 µL cold, filter sterilized PBS, and kept on ice. Propidium iodide (PI) (Invitrogen, Carlsbad, CA, USA) (1 mg/mL solution) was added for 20 min at a final concentration of 30 ng/mL (44.8 µM). Fluorescence was quantified through a BD Accuri C6 Flow cytometer (BD, Franklin Lakes, NJ, USA) and the percent of cells that were PI positive were quantified from a sample of 10,000 events. For the negative control, *P. larvae* cells were prepared in BHIT media and bacteria only. For the positive control, Triton-X100 (0.001% final concentration) was added to culture media and bacteria for 3 h to induce lysis. Four independent biological replicates were performed using three technical replicates for each experiment (*n* = 3).

### 2.6. Quantification of Intracellular Reactive Oxygen Species and Glutathione Levels

To determine if either CAIE, CABE, or CAPE would generate intracellular reactive oxygen species (ROS) in *P. larvae*, cell-permeant 2′,7′-dichlorodihydrofluorescein diacetate (H_2_DCFDA) was used as previously described [36]. Briefly, stock concentrations of each compound (CAIE, CABE, or CAPE) were freshly prepared as 10.5 mg/mL solution in 50% ethanol. *P. larvae* were incubated with 10 µM H_2_DCFDA for 45 min in the dark, washed with PBS, added to the microplate, and treated with either CAIE, CABE, or CAPE at 125 µg/mL or 3% H_2_O_2_ for 18 h at 37 °C. Fluorescence (excitation = 485 nm, emission = 535 nm) was acquired using the Infinite M200 Microplate reader. H_2_O_2_ was added to the cells at a final concentration of 3% and used as a positive control. Three independent biological replicates were performed using two to six technical replicates for each experiment (*n* = 2–6). To quantify intracellular glutathione (GSH) levels, cells were exposed to either CAIE, CABE, or CAPE at their MIC for either 1.5, 3 or 6 h. GSH levels were quantified through the GSH-Glo Glutathione assay as recommended by the manufacturer for cells grown in suspension (Promega, Madison, WI, USA). The number of PL cells were enumerated prior to GSH quantification and 1000 cells were used for each sample. Luminescence was acquired using the Infinite M200 Microplate reader. Luminescence values for all samples were within the GSH standard curve. One independent biological replicate was performed using three technical replicates for each experimental group (*n* = 3). 

### 2.7. Statistical Analysis

Data are presented as mean ± standard deviation (SD). Analyses were done using GraphPad Prism 6.0 (San Diego, CA). A two-way Analysis of variance (ANOVA) was used to compare among more than two experimental groups with two independent variables. A one-way ANOVA was used to compare among more than two experimental groups with one independent variable. A *t*-test was used to compare two groups with one independent variable. Significance was noted at *p* < 0.05.

## 3. Results

The synthesis of the library of caffeic acid esters (Scheme 1) began with an esterification of the synthetically derived caffeic acid diacetate [37]. This was accomplished through the in-situ formation of an acyl chloride with subsequent displacement by an alcohol. The resultant protected esters could be isolated, after column chromatography, in moderate to good yields (41–89%). Deprotection using basic methanol then afforded the pure desired products in moderate yield (33–79%) without the need for further chromatographic purification as observed previously [38]. This simple three-step protocol, originating from commercially available material, allowed for the generation of products on a gram scale in moderate to good (29–68%) overall yield for testing. In all cases, yields were determined by spectroscopically (^1^H-, ^13^C-NMR, FTIR) homogenous material that matched previously reported structural data for these compounds [39].

To determine the MICs of the six caffeic/ferulic acid derivatives, all stock concentrations were solubilized in DMSO and then diluted in BHIT or MYPGP. The MICs obtained were determined after 18 h. The compounds with the lowest MICs were CAIE, CABE, or CAPE, which were 125 µg/mL (Table 1). Three compounds CA, FA, and CACE had MICs greater than 500 µg/mL, which was the highest concentration used in the assay. Since CAIE, CABE, or CAPE were all inhibitory against *P. larvae*, we tested whether a lower MIC could be acquired using the three compounds in a mixture. The three compounds individually had lower MIC’s of 31.25 µg/mL when incubated together with *P. larvae* (Table 1). MBCs were also determined for each effective compound and caffeic acid mixture. The MBCs were equivalent to the MICs (Table 1).

The mechanism of action of these caffeic acid derivatives are currently unknown. To identify whether these compounds acted as bacteriolytic or bacteriostatic compounds, CAIE, CABE, or CAPE were incubated with *P. larvae* at their MICs individually for 18 h. Lysed cells, indicative of a bacteriolytic mechanism, were quantified through flow cytometry and identified as PI positive cells. In the negative control, approximately 7% of cells were PI positive after 18 h, indicating the culture had reached stationary phase (Figure 1). Incubating *P. larvae* in TX-100 for three hours significantly increased the percent PI positive cells to 23% of the total cells enumerated. Incubation with either CAIE, CABE, or CAPE led to a significant reduction in the percent of PI positive cells to less than one percent in all cultures tested, indicating these compounds inhibited bacterial growth without leading to cell lysis.

We hypothesized that the caffeic acid derivatives lead to increased oxidation within the cell. The catechol functional group found in this class of molecules can rapidly undergo a variety of redox processes to form semiquinone radicals and ortho-benzoquinones, both of which can undergo subsequent oxidative processes that can damage and/or destroy DNA and protein structures [40]. In many cases, catechols damage biomolecules through the generation of reactive oxygen species (ROS) such as superoxide (O_2_^−^), hydrogen peroxide (H_2_O_2_), and hydroxyl radicals (OH) [41]. To test this hypothesis, *P. larvae* cells were incubated in H_2_DCFDA, which is a general oxidative stress indicator that is non-fluorescent within cells (H_2_DCFDA) and becomes a highly fluorescent molecule (2′,7′-dichlorofluorescein (DCF)) upon cleavage of the acetate groups by intracellular esterases and oxidation. Incubation of *P. larvae* with CAIE, CABE, or CAPE above or at the MIC (500, 250, and 125 µg/mL) produced a significant increase in fluorescence in a dose response manner, indicating an increase in oxidative stress within cells at all concentrations used (Figure 2). Interestingly, the highest concentrations of CAIE, CABE, or CAPE at 500 µg/mL increased the intracellular fluorescence to a similar degree as 3% H_2_O_2_, which is a well-known oxidant and the positive control. 

To identify whether increased ROS production resulted in a significant change in antioxidant levels, intracellular glutathione (GSH) levels were quantified in *P. larvae* cells during three acute time points (1.5, 3 and 6 h) (Figure 3). GSH levels remained consistent in vehicle treated control cells at all time points indicating the addition of the vehicle (DMSO) had no effect on intercellular GSH levels. However, brief exposure of CAPE and CAIE significantly increased GSH levels in cells by more than 2-fold (0.25 µM to > 0.69 µM) in less than 2 h. GSH levels in PL cells changed most significantly when exposed to CAPE over the 6-h period. After 1.5 h, CAPE generated the highest intracellular GSH levels (0.93 µM), however, after 6 h CAPE decreased GSH levels to the lowest concentration quantified (0.18 µM), indicating that the overproduction of ROS may lead to a subsequent decrease in antioxidant levels. Together these data support the hypothesis that caffeic acid derivatives significantly alter antioxidants levels early during exposure, and act as oxidants within cells at later time points to inhibit binary fission.

## 4. Discussion

As a substance, propolis represents a vast wealth of botanically derived, biologically active, natural products [22]. Despite this, isolating and testing significant quantities of any one component from a propolis matrix is problematic, as purification procedures are complex, and molecules are often found in very low concentrations [29]. Therefore, synthetically generating these natural products on a large-scale is a viable alternative for the testing and development of propolis-based therapies. To this end, we developed a rapid, three-step protocol to prepare the caffeic esters of interest. While methods for a one-step, direct esterification of the caffeic acid parent molecule exist [42], we found that our target molecules were obtained in unsatisfactory yields due to the formation of side products. To prevent these cross-esterification byproducts, we first protected the catechol alcohol functional groups [37]. After esterification of the free acid and subsequent purification, we could deprotect the catechol moiety in good yields over three steps. Conceivably, this procedure could be applied to a wide variety of structurally diverse caffeic acid esters. 

Numerous studies have identified the efficacy and quantified the MICs of different naturally and synthetically derived propolis compounds against Gram-positive, Gram negative bacteria, as well as *P. larvae* [32,43,44,45,46]. Cinnamic acids, such as caffeic acid and ferulic acid, have been studied for their antimicrobial activities and typically show weak growth inhibition against Gram-negative bacteria compared to Gram-positive bacteria [31]. Some phenolic butyl and methyl esters have limited efficacy and inhibit bacterial growth at significantly higher concentrations (1 mg/mL) than the MICs reported in this study [43]. The inhibitory effects of these esters are improved by increasing the length of the alkyl chain to make them structurally similar to the most effective compounds identified in this study (CAPE, CAIE, and CABE). Other studies have identified the increased effectiveness of caffeic acid esters (MIC < 1.25 mM) against bacterial growth compared to simple phenolic esters (MIC > 5–10 mM) [46]. CAPE, derived from propolis from Mexico, was previously shown to be effective against Gram-positive bacteria, specifically *S. aureus*, at concentrations between 100 and 400 µM, which correlates well to the MIC of CAPE against *P. larvae* reported here (440 µM). Interestingly, the combination of CAPE, CAIE, and CABE was a more potent inhibitor of bacterial growth (MIC = ~90 µM) than individual compounds alone indicating an additive interaction of these compounds [32].

Recently, studies looking to identify bioactive constituents of propolis samples from Bulgaria have also identified the caffeic and ferulic ester family of molecules to be candidate therapies against *P. larvae* [33]. Through a series of chemical isolations and subsequent bioassays the researchers identified an inseparable mixture of caffeate esters that were particularly efficacious against the pathogen (lowest MIC reported for caffeate mixture by Bilikova et al. was 31.25 µM after 12 h only). The mixture was tentatively identified as CAPE, CAIE, CABE, and isopentyl caffeic ester. Although in this study the precise concentrations of each component within the mixture was not reported, nor were individual components isolated/tested, this MIC value correlates extremely well with the value determined from the mixture of CAPE, CAIE, and CABE reported here (MIC = 31.25 µM). 

Interestingly, in our study, CACE, while structurally similar to CAPE, CAIE, and CABE, proved to be considerably less effective (MIC > 500 µM). Correspondingly, CACE was not identified as an active component in the abovementioned fractionated bioassay study [33]. This indicates that subtle structural differences may play a significant role in the activity of this family of compounds against *P. larvae*. 

Although many studies have identified the inhibitory effects of phenolic esters and acids, few studies have been successful at identifying a potential mechanism of action for these antimicrobials [31]. Two main mechanisms of action have been characterized for caffeic acid esters including disruption of membrane permeability and alteration of redox potentials in cells. For example, a previous study showed that 4-hydroxycinnamic acid, a cinnamic acid derivative, disrupted the outer membrane of the Gram-negative bacteria increasing the permeabilization causing cytoplasmic leakage [31]. CAPE was also shown to inhibit the growth of Gram-positive bacteria, such as *Bacillus*, by disrupting membrane permeability through both a bacteriostatic and bacteriocidic effect depending on the duration of treatment [32]. However, certain caffeic alkyl esters induce a different mechanism of action through altering the oxidant/antioxidant balance within cells leading to cell death and decreased viability [46]. Specifically, CAPE has been shown to alter oxidative processes and induce apoptosis in human cells. CAPE therefore may act through a redox mechanism as a pro-oxidant since apoptosis induced by CAPE was significantly reduced with *N*-acetyl-l-cysteine in human cells [47]. 

Our data support the hypothesis that CAPE, CABE, and CAIE all inhibit PL growth and kill PL cells without leading to cell lysis when used at their MIC for 18 h (Table 1 and Figure 1). The untreated PL cells reached stationary phase after 18 h of growth, as indicated by less than 10% of cells, which were PI positive. In contrast, in the presence of a low concentration of detergent (TX-100 at 0.001%) PL cells had a small but significant increase in the number of permeabilized (or lysed) cells, which could be accurately quantified. We hypothesize the low number of permeabilized PL cells observed incubated with CAPE, CABE and CAIE is due to an inhibition and killing of PL cells that resulted in fewer total cells in culture and even fewer cells that have disrupted cell walls. Interestingly, a previous study suggested that CAPE leads to bacterial cell lysis or a bacteriocidic effect, after incubation times of greater than 6 h, and these effects are not the same during acute incubation times where a bacteriostatic mechanism occurs. This inconsistency may be due to the different concentrations and bacteria used between studies [32]. However, CAPE can have differential effects within cells that may influence the cellular phenotypes observed. For example, CAPE can act as an antioxidant in cells, but also possesses oxidant activity [31]. Our data support this dichotomous hypothesis because the most effective caffeic acid esters induced changes in oxidant levels in bacterial cells as determined by the significant formation of ROS after 18 h, and significant changes in antioxidant intracellular GSH levels during acute (<6 h) time periods which was unexpected. Increased ROS production can lead to oxidative DNA damage most likely due to the production of free radicals, such as superoxide or the hydroxyl radical, which can directly oxidize DNA into lesions like 8-hydroxy-2ʹ-deoxyguanosine (8-oxo-2dG) [48] interfering with DNA replication, which more likely leads to a bacteriostatic effect than a bacteriocidic effect [49]. Alternatively, Gram-positive bacteria, such as *B.* subtilis, can alter expression levels of glutathione associated genes in response to oxidative stress [50] or induce the expression of regulons such as OxyR and SoxRS [51].

## 5. Conclusions

In conclusion, we have been able to synthetically prepare and test a series of caffeic/ferulic acids and esters against PL. We have shown that a subset of our compound library, CAPE, CAIE, and CABE inhibit the growth of PL and act as bactericidal compounds. Further experimentation indicates that these compounds alter the oxidant and antioxidant balance in cells through ROS generation and acute GSH fluctuations. We were also able to show that small structural changes within the compound library can have large effects on MIC. These results are significant because CAPE, CAIE, and CABE represent a small group of naturally occurring compounds that could be used in the future to treat antibiotic resistant PL strains. Ultimately, further screening of this family of molecules may lead to significant improvements in antimicrobial efficacy and the potential for the development of novel therapies for the prevention of AFB.

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
