# Peer review of "Caffeic Acid Esters Are Effective Bactericidal Compounds Against *Paenibacillus"

_biomolecules, 2019, doi:10.3390/biom9080312_

Round 1

Reviewer 1 Report

This manuscript reports on the synthesis and anti-Paenibacillus larvae action of synthetically obtained natural caffeic acid esters. The mechanism of action of the synthesized compounds was also studied. The methods applied are up-to-date and appropriate, the conclusions are supported by the results. The results are valuable. There are however some minor points which need additional attention by the authors:

1.       Page 2, line 54: antibiotics against AFB are banned in the EU.

2.       In Materials and Methods, please add the yield of synthesis of diacetylcaffeic acid

3.       The authors should clearly explain what the advantages of their method are in comparison to other synthetic approaches, (described recently e.g. in Bankova et al., Comptes rendus de l'Academie bulgare des Sciences 71(9), 1157 – 1166 (2018)?

4.       What is the overall yield based on the starting material caffeic acid?

5.       Did the authors observe any traces of the Z-isomers in NMR spectra?

Author Response

Response to Reviewer 1’s comments:

1.       Page 2, line 54: antibiotics against AFB are banned in the EU.

            This line has been changed to emphasize this ban within the EU.

2.       In Materials and Methods, please add the yield of synthesis of diacetylcaffeic acid

            This change has been made.

3.       The authors should clearly explain what the advantages of their method are in comparison to other synthetic approaches, (described recently e.g. in Bankova et al., Comptes rendus de l'Academie bulgare des Sciences 71(9), 1157 – 1166 (2018)?

            The advantage of this method (no need for challenging purification) over previous methods has been inserted and this citation has been added.

4.       What is the overall yield based on the starting material caffeic acid?

            The overall yield range for the four syntheses (29-68% over 4 steps each) has been added.

5.       Did the authors observe any traces of the Z-isomers in NMR spectra?

            The commercially available starting material was the geometrically pure caffeic acid, and thus no isomers were observed at any point in the synthesis.

Reviewer 2 Report

Lines 15-16- make it clearer in the line 15 that propolis can contain caffeic acid esters.

Line 37- not all parasites, like Varroa destructor mites, “infect” honey bees

Line 45- “larva” or “larvae”? I would think “larvae” is more appropriate.

Line 51-Tylosin tartrate/Tylan is also used to treat American foulbrood. Although it would be considered an off-label treatment, many beekeepers will use Tylan prophylactically.

Line 65- As far as I understand, the metric “molecules per sample” has no meaning since “sample” is not a defined unit. Please clarify what you mean by “sample”.

Line 73-74- Can you list any plants from which honey bees may collect resin containing caffeic esters? Since you mention there is great variability in propolis composition across season, location, and plant availability, I think it would be great to offer the audience some context to the relevance of focusing on caffeic esters.

Line 75- Gram-positive should be capitalized, since it was named after a person.

Line 189- Why were two different liquid broths used? Couldn’t this potentially confound your MIC results? Please explain why using two different broths did not confound your results or what you did to account for this. I see later on you mention it briefly in lines 196-197, but please provide more detail how you arrived to this assessment.

Line 199- Italicize P. larvae.

Line 421- Change ABF to AFB.  

Table 1- It looks like the table was not formatting correctly, as there are odd symbols in the middle of the text. Please fix.

Author Response

Response to Reviewer 2’s comments:

Lines 15-16- make it clearer in the line 15 that propolis can contain caffeic acid esters.

            This change has been made.

Line 37- not all parasites, like Varroa destructor mites, “infect” honey bees

            This sentence has been modified to pest and pathogens doing “harm” rather than infecting honey bees.

Line 45- “larva” or “larvae”? I would think “larvae” is more appropriate.

            This change has been made.

Line 51-Tylosin tartrate/Tylan is also used to treat American foulbrood. Although it would be considered an off-label treatment, many beekeepers will use Tylan prophylactically.

            This has been added to the sentence.

Line 65- As far as I understand, the metric “molecules per sample” has no meaning since “sample” is not a defined unit. Please clarify what you mean by “sample”.

            This sentence has been clarified to explain that up to 300 molecules are found within each resin sample that is analyzed.

Line 73-74- Can you list any plants from which honey bees may collect resin containing caffeic esters? Since you mention there is great variability in propolis composition across season, location, and plant availability, I think it would be great to offer the audience some context to the relevance of focusing on caffeic esters.

 This change has been made.

Line 75- Gram-positive should be capitalized, since it was named after a person.

This suggestion has been included in the revision.

Line 189- Why were two different liquid broths used? Couldn’t this potentially confound your MIC results? Please explain why using two different broths did not confound your results or what you did to account for this. I see later on you mention it briefly in lines 195-197, but please provide more detail how you arrived to this assessment.

The reasoning behind using two different culture media have been further emphasized and included in in lines 191-193 in the revised manuscript.

Line 199- Italicize P. larvae.

This suggestion has been included in the revision. 

Line 421- Change ABF to AFB.  

This suggestion has been included in the revision.

Table 1- It looks like the table was not formatting correctly, as there are odd symbols in the middle of the text. Please fix.

Table 1 has been included as a PDF file in the revision.

Reviewer 3 Report

Thank you for the opportunity to review the manuscript by Collins et al. titled, “Caffeic acid esters are effective bacteriostatic compounds against Paenibacillus larvae by altering intracellular oxidant and antioxidant levels”.

Overview: This study explores new data to describe caffeic acid esters to suppress growth of Paenibacillus larvae, an important bacterial pathogen of honey bees. Given current and global concern over health of honey bees, this topic is relevant. The methods utilized were applied in novel ways in this study, the manuscript is well written, and the authors’ interpretations are generally sound. However, several concerns about the manuscript are provided below for consideration.

Major concerns:

1.       The authors mention the MIC for a combination of caffeic acid esters (such as on line 21) in showing activity against P. larvae. However, it not clear if the MIC corresponds to each of the caffeic acid ester concentrations alone, or the sum of their concentrations. This point should be clarified, as the data suggest either additive or synergistic effects of the substances on the bacterium.

2.       In line 278 the authors indicate that treatment of P. larvae with TX-100 causes permeabilization of only 23 percent of the cells; this unexpected finding should receive comment by the authors. In addition, treatment P. larvae cells by either CAIE, CABE, or CAPE seems to reduce the permeabilization (beginning on line 279) below the value of 7 percent observed for the negative control. This unexpected finding should also receive comment.

3.       The modest signal observed in the analysis of GSH in P. larvae raises some concerns. A closer look reveals:

a.       Some literature suggests glutathione is not produced in most genera of Gram-positive bacteria. According to Fahey et al, glutathione was not found in either of the two Gram-positive endospore-producing bacteria, Bacillus cereus and Bacillus subtilis (Occurrence of Glutathione in Bacteria, JOURNAL OF BACTERIOLOGY, Mar. 1978, p. 1126-1129). Can the authors comment on the likelihood that P. larvae could express glutathione based on other evidence, preexisting literature, or genetic capability?

b.       The Promega GSH-Glo™ Glutathione assay kit appears to be designed primarily for mammalian cells and may not be suitable for evaluating the presence of intracellular glutathione in bacteria that contain cell walls.

c.       The explanation for decreasing GSH levels over time in treated cells (in the sentences beginning in lines 314 and 317) is not compelling. Can a reference be provided for evidence of a similar effect in another system?

d.       The Figure 3 legend states, “Asterisks represent a significant difference between control cells and cells exposed to either CAIE, CAPE, CABE (p < 0.05).” However, an asterisk does not appear above the bar for CABE. 

Minor concerns:

1.       It is not clear why the authors use both ABF and AFB for an abbreviation for American Foulbrood, when AFB is conventionally used.

2.       Capitalization of the word, caffeic within sentences in several places of the manuscript (such as in line 19, …“Caffeic acid benzyl ester (CABE), and Caffeic acid phenethyl ester”… does not appear warranted.

3.       Beginning in line 71 ,the authors state, “To directly address this issue, our laboratories have begun a program in the systematic synthesis and testing of individual natural product components of propolis against PL.” Yet at that point in the manuscript, the authors offer make no reference to the point that caffeic acid esters are present in propolis. This possibility is not referenced until the paragraph that begins on line 352. The connection between propolis and the molecules the authors tested should therefore be made clear and early in the manuscript, preferably beginning in the abstract or introduction.

4.       The units µg/ml and µM to describe the MIC of compounds vs P. larvae lead to some confusion. Adding to this problem, the authors report the MIC for the combination of compounds they tested as both 31.25 µg/ml (various locations in the manuscript) and 31.25 µM (line 375).

5.       On line 41, the authors provide the following phrase, “This extremely contagious bacteria, known commonly in the beekeeping community as American foulbrood (AFB)…”. The authors are assumed to be referring to a single type of organism, therefore the word “bacteria” should be changed to “bacterium”.

6.       In line 184 and at several locations elsewhere, the Latin name Paenibacillus larvae and P. larvae should be italicized.

7.       On line 188, some incubation conditions for P. larvae are provided. However, the authors do not indicate if the culture was rotated or agitated for aeration. This should be clarified, as it can impact the condition of the organism once grown.

8.       The authors confirmed the caffeic acid esters did not lyse the P. larvae cells. However, it is possible the cells were killed without lysis. Therefore, they did not rule out the possibility that treatment resulted in a bactericidal effect. If the result is cell killing rather than inhibition of growth, that point is important and could increase the significance of the findings. It may be worth showing colony forming unit counts on a solid medium before and after treatment to determine whether the effect is bacteriostatic or bactericidal.

Author Response

Responses to Reviewer 3’s comments:

Major concerns:

1.       The authors mention the MIC for a combination of caffeic acid esters (such as on line 21) in showing activity against P. larvae. However, it not clear if the MIC corresponds to each of the caffeic acid ester concentrations alone, or the sum of their concentrations. This point should be clarified, as the data suggest either additive or synergistic effects of the substances on the bacterium.

The MIC given was for individual caffeic acid ester that were used in combination. Therefore, the effect of this combination was additive. This has been clarified in the abstract (line 21) and in discussion (line 376) of the revision. We thank Reviewer 3 for this suggestion.

2.       In line 278 the authors indicate that treatment of P. larvae with TX-100 causes permeabilization of only 23 percent of the cells; this unexpected finding should receive comment by the authors. In addition, treatment P. larvae cells by either CAIE, CABE, or CAPE seems to reduce the permeabilization (beginning on line 279) below the value of 7 percent observed for the negative control. This unexpected finding should also receive comment.

The Triton concentration used as a positive control to permeabilize the PL cells was low (0.001%) and has been included in the revision (line 222). We hypothesize that the low level of permeabilization observed in the PL cells incubated with CAIE, CABE and CAPE was due to an inhibition of binary fission caused by these compounds over the 18 h incubation period. The negative control cells proliferated normally and reached stationary phase, which is consistent with the level of intact and permeabilized cells. Both suggestions have been included in the revised discussion (line 408-414).

3.       The modest signal observed in the analysis of GSH in P. larvae raises some concerns. A closer look reveals:

a.       Some literature suggests glutathione is not produced in most genera of Gram-positive bacteria. According to Fahey et al, glutathione was not found in either of the two Gram-positive endospore-producing bacteria, Bacillus cereus and Bacillus subtilis (Occurrence of Glutathione in Bacteria, JOURNAL OF BACTERIOLOGY, Mar. 1978, p. 1126-1129). Can the authors comment on the likelihood that P. larvae could express glutathione based on other evidence, preexisting literature, or genetic capability?

The complete genome of P. larvae was published in 2017 by Dingman. Currently there are no published data indicating that P. larvae express glutathione associated enzymes. However, genetic responses to oxidative stress occur in bacteria through the SoxRS and OxyR regulons, and occurs in general all organisms capable of aerobic respiration including P larvae (Cabiscol et al. (2000)). The lack of data for glutathione associated genes in P. larvae does not exclude the possibility that P. larvae could synthesize or utilize GSH from the environment. Indeed, although B. subtilis was shown not to express GSH in 1978, a more recent publication in 2012 in FEMS Microbiology Letters by Zhang et al. indicates that B. subtilis express glutathione S-transferase in response to oxidative stress. Therefore, oxidative stress induced may be due to imbalance of redox potential in bacterial cells, which can lead to transcriptional changes in bacterial cells. 

b.       The Promega GSH-Glo™ Glutathione assay kit appears to be designed primarily for mammalian cells and may not be suitable for evaluating the presence of intracellular glutathione in bacteria that contain cell walls.

The authors thank Reviewer 3 for this comment. We believe that the reviewer has a concern about inefficient lysis of the PL bacterial cells or the inaccurate GSH quantification due to inefficient lysing. It is true that this quantification method (GSH-Glo) has not been used extensively in bacterial systems, there have been publications using this method previously published including:

Reduced Glutathione Mediates Resistance to H2S Toxicity in Oral Streptococci

Dysregulation of transition metal ion homeostasis is the molecular basis for cadmium toxicity in Streptococcus pneumoniae

In addition, while quantifying GSH levels from PL cells, the cells were enumerated prior to lysis and only 1000 cells were used for each sample, which was the lowest number of cells suggested by the manufacturer. This information has been included in the methods section (line 238-239).

c.       The explanation for decreasing GSH levels over time in treated cells (in the sentences beginning in lines 314 and 317) is not compelling. Can a reference be provided for evidence of a similar effect in another system?

As described in the response above for comment a., the paper from 2012 in FEMS Microbiology Letters by Zhang et al. that indicates B. subtilis express glutathione S-transferase in response to oxidative stress. This example and a review describing the transcriptional expression of multiple antioxidant operons have been included in the revision (lines 428 to 430).

d.       The Figure 3 legend states, “Asterisks represent a significant difference between control cells and cells exposed to either CAIE, CAPE, CABE (p < 0.05).” However, an asterisk does not appear above the bar for CABE. 

The authors thank Reviewer 3 for this insightful comment. The GSH levels of PL cells treated with CABE were not significantly different than the vehicle control after 1.5 hours. The figure legend has been modified to reflect this change.

Minor concerns:­

1.       It is not clear why the authors use both ABF and AFB for an abbreviation for American Foulbrood, when AFB is conventionally used.

The authors thank Reviewer 3 for this important edit. Lines 12, 13, 60 and 424 have be revised to indicate American Foulbrood as AFB and not ABF. We are sincerely sorry for this mistake.

2.       Capitalization of the word, caffeic within sentences in several places of the manuscript (such as in line 19, …“Caffeic acid benzyl ester (CABE), and Caffeic acid phenethyl ester”… does not appear warranted.

            This change has been made.

3.       Beginning in line 71 ,the authors state, “To directly address this issue, our laboratories have begun a program in the systematic synthesis and testing of individual natural product components of propolis against PL.” Yet at that point in the manuscript, the authors offer make no reference to the point that caffeic acid esters are present in propolis. This possibility is not referenced until the paragraph that begins on line 352. The connection between propolis and the molecules the authors tested should therefore be made clear and early in the manuscript, preferably beginning in the abstract or introduction.

            This change has been made – the connection is now made in the abstract.

4.       The units µg/ml and µM to describe the MIC of compounds vs P. larvae lead to some confusion. Adding to this problem, the authors report the MIC for the combination of compounds they tested as both 31.25 µg/ml (various locations in the manuscript) and 31.25 µM (line 375).

The authors are sorry for the confusion including two different units for concentrations. Both µg/ml and µM were included in the manuscript because previous published results using P. larvae quantify MIC’s using both units (µg/ml and µM). Therefore, the current manuscript attempted to compare equivalent units from previous studies to determine if CAPE, CABE and CAIE were as effective as previously described compounds. The use of µM has been clarified in the revised manuscript.

5.       On line 41, the authors provide the following phrase, “This extremely contagious bacteria, known commonly in the beekeeping community as American foulbrood (AFB)…”. The authors are assumed to be referring to a single type of organism, therefore the word “bacteria” should be changed to “bacterium”.

This suggestion has been included in the revision.

6.       In line 184 and at several locations elsewhere, the Latin name Paenibacillus larvae and P. larvae should be italicized.

This suggestion has been included in the revision.

7.       On line 188, some incubation conditions for P. larvae are provided. However, the authors do not indicate if the culture was rotated or agitated for aeration. This should be clarified, as it can impact the condition of the organism once grown.

This suggestion has been included in the revision in line 189.

8.       The authors confirmed the caffeic acid esters did not lyse the P. larvae cells. However, it is possible the cells were killed without lysis. Therefore, they did not rule out the possibility that treatment resulted in a bactericidal effect. If the result is cell killing rather than inhibition of growth, that point is important and could increase the significance of the findings. It may be worth showing colony forming unit counts on a solid medium before and after treatment to determine whether the effect is bacteriostatic or bactericidal.

The authors thanks Reviewer 3 for this excellent suggestion. Quantifying the minimum bactericidal concentration (MBC) would provide additional data for the paper. If the acceptance of this paper is contingent upon quantifying the MBC, the authors would like to request a two to three week extension from the 5 day limit in order to correctly and completely address this concern.